# Demonstration of non-destructive and isotope-sensitive material analysis using a short-pulsed laser-driven epi-thermal neutron source

Marc Zimmer [1✉], Stefan Scheuren [1], Annika Kleinschmidt[2,3], Nikodem Mitura[1], Alexandra Tebartz[1], Gabriel Schaumann[1], Torsten Abel[1], Tina Ebert [1], Markus Hesse [1], Şêro Zähter[2], Sven C. Vogel [4], Oliver Merle[5], Rolf-Jürgen Ahlers[5], Serge Duarte Pinto [6], Maximilian Peschke[7], Thorsten Kröll [1], Vincent Bagnoud [2], Christian Rödel[1] & Markus Roth[1]

Neutrons are a valuable tool for non-destructive material investigation as their interaction cross sections with matter are isotope sensitive and can be used complementary to x-rays. So far, most neutron applications have been limited to large-scale facilities such as nuclear research reactors, spallation sources, and accelerator-driven neutron sources. Here we show the design and optimization of a laser-driven neutron source in the epi-thermal and thermal energy range, which is used for non-invasive material analysis. Neutron resonance spectroscopy, neutron radiography, and neutron resonance imaging with moderated neutrons are demonstrated for investigating samples in terms of isotope composition and thickness. The experimental results encourage applications in non-destructive and isotope-sensitive material analysis and pave the way for compact laser-driven neutron sources with high application potential.

[1] Technische Universität Darmstadt, Institut für Kernphysik, Darmstadt 64289, Germany. [2] GSI Helmholtzzentrum für Schwerionenforschung, Darmstadt 64291, Germany. [3] Helmholtz Institut Jena, Jena 07743, Germany. [4] Materials Science and Technology Division, Los Alamos National Laboratory, Los Alamos, NM 87545, USA. [5] ProxiVision GmbH, Bensheim 64625, Germany. [6] Photonis Netherlands, B.V., Roden 9301 ZR, The Netherlands. [7] Surface Concept GmbH, Mainz 55124, Germany. ✉email: mzimmer@ikp.tu-darmstadt.de

Since the discovery of neutrons in 1932[1], neutron sources have evolved into a valuable diagnostic tool for science and industry. As neutrons interact dominantly via the strong force with the nucleus when propagating through matter, they possess high penetration depths and their interaction probability is determined by the nucleus configuration. The attenuation of a neutron beam $I = I_0 \exp(-\Sigma \cdot d)$ inside a material thus depends on the macroscopic cross-section $\Sigma$, which is isotope and energy-specific, as well as on the material thickness $d$[2]. Structural information of samples can be obtained by measuring the attenuation with spatially resolving detectors—typically referred to as neutron radiography[3,4]. As the neutron cross-sections have strong isotope-dependent resonances in the so-called epithermal energy regime between 1 eV and several keV, energy-sensitive measurements enable isotope-selective material characterization. This is pursued in neutron resonance spectroscopy (NRS), where the attenuation of neutrons through a sample is measured energy-resolved, using the time-of-flight of a short-pulsed neutron source[5]. When spectroscopic and spatially resolving techniques are combined, this is typically referred to as neutron resonance imaging (NRI), which can be used for investigating the material and isotope distribution of a sample[6]. Due to typical flight times of epithermal neutrons in the range of tens of μs, the neutron source has to be short-pulsed. So far, the requirements on the neutron flux and temporal pulse width limit the use of these techniques to large-scale accelerator-based neutron sources such as the Los Alamos Neutron Science Center[7], which consists of a 800-m proton linear accelerator, or GELINA, a neutron time-of-flight facility based on an electron accelerator[8]. The large size and investment cost of these neutron sources limit their availability and access for users. With the development of laser systems with petawatt peak power[9], laser-driven proton accelerators[10] facilitate comparably compact neutron sources to be build in the range of meters instead of kilometers with initial pulse durations in the range of nanoseconds[11–17]. Laser-driven neutron sources (LDNS) have been proposed for NRS and NRI[18,19] but despite the large progress in the field of laser-particle acceleration in the last years, laser-driven neutron sources were mainly limited to neutron radiography using fast neutrons[11,20]. LDNS with thermal and epithermal energies would enable the investigation of samples with material and isotope-specific contrast by exploiting the sharp resonances in the thermal and epithermal range[21].

In this article, we present an optimized LDNS in the epithermal and thermal energy range. The design, optimization, and characterization of the LDNS is described in detail. We show experimental results and demonstrate NRS using epithermal neutrons, neutron radiography using thermal neutrons, and a simplified variant of NRI with thermal neutrons. With NRS, a sample made of tungsten is investigated as a proof-of-principle. Three different isotopes of tungsten, as well as a tantalum impurity, are identified. With thermal neutron radiography, we measured the thickness and position of samples through lead shielding. A proof-of-concept of NRI was applied to identify cadmium in a sample arrangement by scanning through resonant and non-resonant energy ranges.

## Results

The experiment was performed using the PHELIX high-intensity laser at the GSI Helmholtz Centre for Heavy Ion Research[22], which can provide one laser pulse every 90 min. A schematic illustration of a laser-driven epithermal neutron source is shown in Fig. 1a. A 3D drawing of the experimental setup at the PHELIX laser is shown, true to scale, in Fig. 1b. In the experiments, laser pulses with an average pulse energy of $103 \pm 9$ J and a pulse duration of 600 fs were focused to a focal spot diameter of $4.0 \pm 0.5$ μm

(FWHM). This leads to a peak intensity of $(2.0 \pm 0.3) \cdot 10^{20}$ W/cm². Deuterated and non-deuterated polystyrene foils with a thickness ranging from 400 to 1100 nm were used as targets for the acceleration of light ions. Protons and deuterons were accelerated in the regime of Target Normal Sheath Acceleration (TNSA)[23] with an exponential spectrum and average cut-off energies of $37 \pm 13$ MeV for protons (p) and $27 \pm 9$ MeV for deuterons (d). Differences in the target thickness had no significant effect on the proton and deuteron yield, which are an indication of the good temporal contrast of the laser system[24].

Laser-accelerated protons and deuterons are directed into a converter material made of beryllium (Be), lithium fluoride (LiF) or a combination of both (LiF-Be). Inside the converter, neutrons are produced via (p,n) and (d,n) nuclear reactions[25] creating an MeV neutron beam with $(1.6 \pm 0.4) \cdot 10^{10}$ neutrons per shot (Fig. 1c, d), in good agreement with previous experiments[13]. The conversion efficiency from protons to neutrons is estimated to $\eta_{conv} = 8 \cdot 10^{-4}$ for our experimental setup based on PHITS Monte Carlo simulations. Fast neutrons with MeV energies are produced in all directions, but with a strong component in forward direction[11]. The primary fast neutrons are measured with different fast detectors, which can discriminate the neutron signal from the gamma signal by the time-of-flight.

The fast neutrons in forward direction are then caught by a moderator made of polyethylene, where the neutrons reduce their energy through elastic collisions with the protons of the moderator material. Moderated neutrons then propagate via diffusion to the moderator surface where they are emitted in all directions. A fraction of the pulsed neutrons from the moderator propagate the time-of-flight distance to the detectors. Samples can be placed in front of the detectors so that the moderated neutrons are measured in transmission of the sample. An optimized collimation system is therefore essential in order to prevent scattered neutrons from interfering with the measurement.

**Optimization of the laser-driven neutron source**. We use the Monte Carlo code PHITS[26] for optimizing the neutron production, moderation, collimation, and detection for our setup. In a first optimization step, an effective converter was designed to achieve a high conversion efficiency of ions into neutrons. Lithium and beryllium are suitable converter materials for LDNS due to their low stopping power and high (x,n) cross-sections[13,27–29]. While lithium has the highest neutron yield for proton energies below 15 MeV, beryllium has a larger neutron yield for higher energies[2]. The exponential proton and deuteron spectrum by TNSA contains energies from both regions. We therefore investigated experimentally, which converter material produces the highest neutron yield. Figure 1c and d shows the neutron flux measured with a fast scintillator and photomultipliers using different converters made of LiF, Be and LiF-Be (Methods). The shaded area marks the standard deviation, the solid lines are the average flux of multiple shots. The scintillator was placed at a distance of 3.4 m at an angle of $38° \pm 2°$ with respect to the target normal direction. Our measurements have shown that the combination of deuterated targets and the LiF converter produced the highest number of neutrons with $(1.1 \pm 0.6) \cdot 10^9$ n/sr and energies up to $83 \pm 5$ MeV. The combination of deuterated targets and beryllium produced an integrated neutron yield of $(1.7 \pm 1.2) \cdot 10^8$ n/sr. The combination of a non-deuterated target on LiF resulted in $(2.0 \pm 1.0) \cdot 10^8$ n/sr.

The neutron yield was further investigated with bubble detectors, which can measure the fast neutron flux while being insensitive to gamma radiation[30]. These detectors were placed around the target chamber. The average fast neutron flux measured by the bubble detectors is shown in Fig. 1e (blue bars)

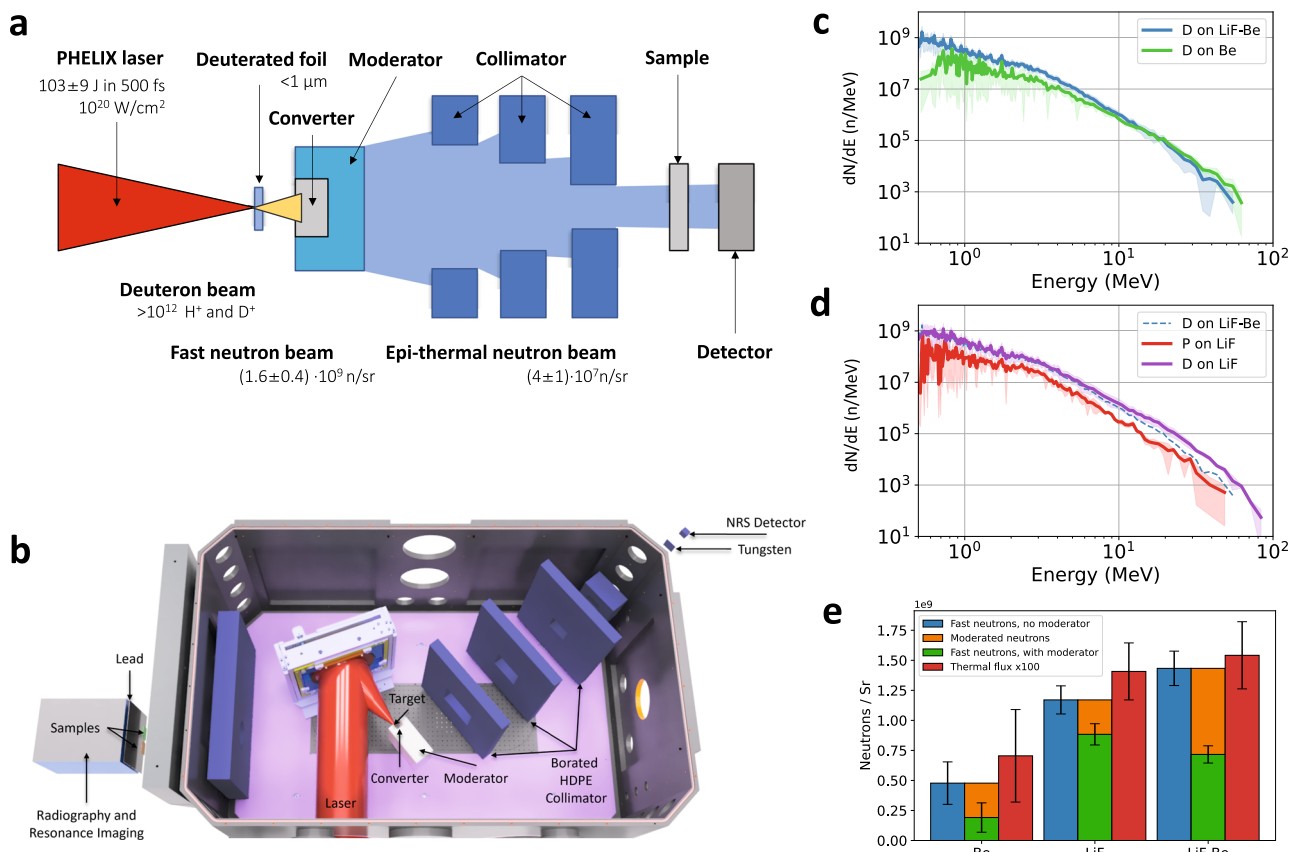

**Fig. 1 Optimization of the fast neutron flux from laser-driven neutron sources (LDNS). a** Schematic setup of an LDNS that produces fast neutrons by the conversion of laser-accelerated protons and deuterons in a Be, LiF, or a LiF-Be converter material. Fast neutrons are subsequently moderated to thermal and epithermal energies. The laser-driven neutron source can be applied for investigating samples with neutron radiography, neutron resonance spectroscopy (NRS), and neutron resonance imaging (NRI). **b** Experimental setup at the PHELIX laser. The moderated neutrons can be simultaneously utilized for an NRS measurement, neutron radiography or an NRI measurement with samples placed outside of the chamber. **c**, **d** Fast neutron spectra measured with a fast scintillator and a photomultiplier for different target and converter combinations. The highest fast neutron flux is obtained for deuterated targets and the LiF converter. **e** Neutrons per steradian are measured for different converter materials. The fast neutron flux is measured with bubble detectors (blue, green). The thermal and epithermal neutron flux is measured with a borated MCP (red). The error bars indicate the standard deviation of the dataset. The orange bar displays the difference between shots with and without moderator and is an estimate of the moderated neutron yield.

for different converter materials. Error bars shown are the statistical standard deviation. The combination LiF-Be provides the highest total neutron yield of $(1.43 \pm 0.14) \cdot 10^9$ n/sr in agreement with the measurements with the fast scintillators. The green bars show the average neutron flux of all bubble detectors with the moderator in place. Accordingly, the difference between those two measurements indicates, which fraction of neutrons is moderated (orange bars). The largest difference is observed with the LiF-Be converter. The thermal neutron flux was additionally measured with a borated MCP (red bars). We observed that the combination LiF-Be produces the highest amount of thermalized neutrons. This can be explained by a lower average kinetic energy of neutrons for the LiF-Be converter compared to LiF, which results in a more efficient moderation. For this reason, the LiF-Be combination was favored for the experiment.

**Moderator optimization**. The moderator was designed to maximize the epithermal neutron flux for energies between 1 and 10 eV at the detector position. We placed the NRS detector perpendicular to the direction of the fast electrons and ions as shown in Fig. 1b. This configuration minimizes the background from gammas and fast neutrons at the position of the NRS detector and reduces the so-called gamma flash in comparison to the neutron

signal. We therefore have to optimize the flux of moderated neutrons, which are emitted from the side of the moderator. PHITS simulations were performed to find the optimal moderator dimensions. In the simulations, the cylindrical beryllium converter was surrounded by a cuboid of polyethylene (Fig. 2a). The epithermal flux between 1 and 10 eV was simulated at a distance of 50 cm perpendicular to the target normal direction. The dimensions of the cuboid was varied in the simulations to optimize the moderated neutron flux (Fig. 2b–d). Error bars are the result of limited statistics. The flux increases for moderator lengths up to 12 cm and enters a regime of saturation. A similar behavior is seen for changes in height with a saturation in flux above 20 cm. An increase in width first leads to a rise in neutron flux as more neutrons are moderated until a width of 8 cm. Afterward, the flux is reduced as less neutrons reach the surface via diffusion from the moderator center. We further added two additional wings with a size of $3 \times 3 \times 20$ cm³ at the laser-facing side, which enhance the flux between 40% at 1 eV and 60% at 10 eV (Fig. 2f). In this configuration, the second peak from d increases, leading to a new maximum at a width of 10 cm. Based on these simulations, the moderator was developed with a dimension of $17 \times 20 \times 10$ cm³. The length was set to 17 cm for reasons of radiation safety but could be reduced to 12 cm without flux degradation. Figure 2e shows the simulated epithermal

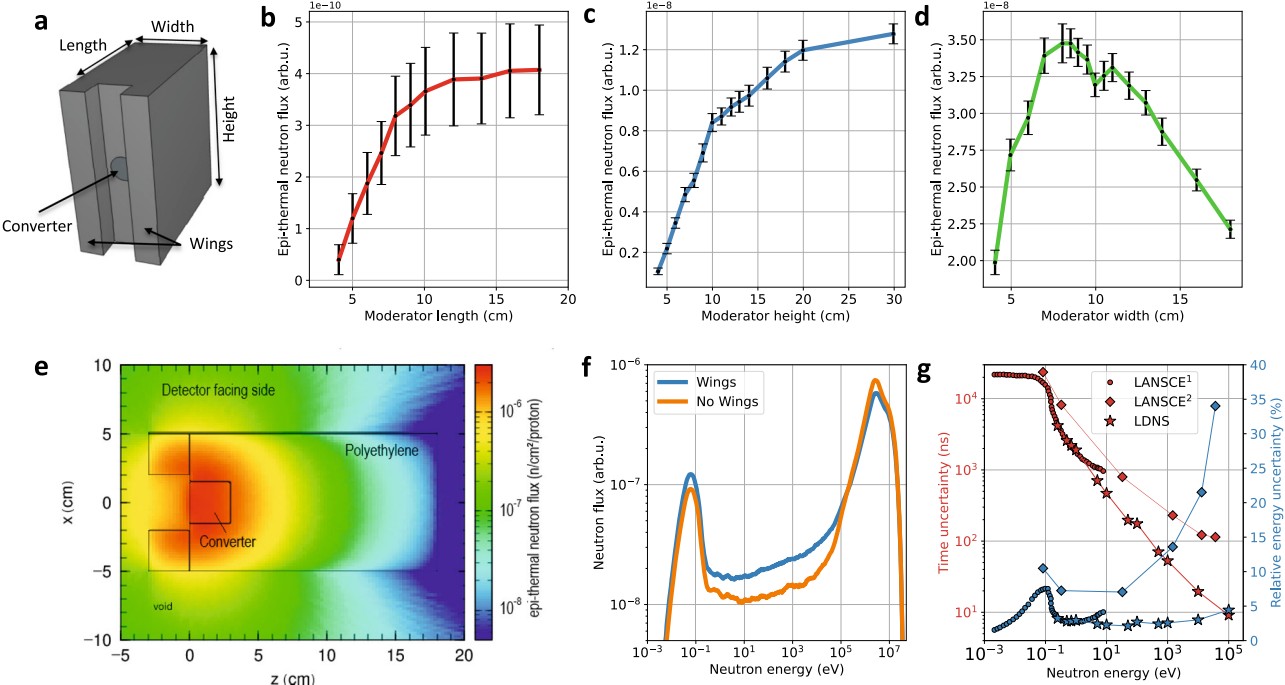

**Fig. 2 Optimization of the epithermal neutron flux from the moderator. a** Sketch of the moderator geometry. **b–d** Epithermal (1–10 eV) neutron flux from a PHITS simulation at the detector position as a function of moderator dimensions. A moderator with $17 \times 20 \times 10$ cm³ was chosen for the experiment. Error bars are the result of statistical uncertainties in the simulation and represent one standard deviation. **e** Epithermal flux distribution in the horizontal plane simulated with the measured neutron flux from Fig. 1d (p on LiF). Most epithermal neutrons are emitted in the vicinity of the converter. **f** Simulated neutron flux at the detector with and without the moderating wings. The wings drastically improve moderation efficiency. **g** (red) Simulated ToF uncertainty caused by the moderation for this LDNS moderator design in comparison to simulations for the LANSCE[31,32] spallation source. (blue) Resulting energy uncertainty at 1.8 m distance for both source types. Moderated LDNS have shorter pulse lengths than conventional neutron sources enabling shorter ToF distances for the same energy resolution.

neutron flux across the moderator cross-section (top view). The simulation shows that most epithermal neutrons are produced in the vicinity of the converter. The moderation efficiency was measured to ~2.5 %, which is about one order of magnitude higher than the results reported by Mirfayzi et al.[17].

**Evaluation of energy resolution.** The collisions of neutrons in the moderator lead to a strongly increased neutron pulse width, which becomes—together with the position of the last collision inside the moderator—the main uncertainty for the time-of-flight measurements. This so-called surface crossing time was simulated using PHITS. The resulting time uncertainty (FWHM) for different energies is displayed in Fig. 2g (red stars). This time uncertainty leads to a relative energy uncertainty at the detector distance of 1.8 m for our NRS measurement (blue stars). For comparison, two datasets from the LANSCE facility are shown. The first dataset, LANSCE1[31], reported the TOF pulse width (red circles). The second dataset, LANSCE2[32], reported the energy uncertainty at 7.93 m and was first calculated into a ToF uncertainty (red diamonds). The TOF pulse width was then used to calculate a relative energy uncertainty at 1.8 m (blue diamonds). A comparison of LANSCE and the LDNS shows that both neutron sources have similar pulse widths below 1 eV. In this region, the time uncertainty is dominated by the moderation time and diffusion towards the moderator surface. For higher energies, however, the time uncertainty for the LDNS decreases faster than for LANSCE. This can be explained by the shorter ion pulse width of the LDNS (<1 ns) in comparison to the 125 ns at LANSCE[33]. While the relative energy uncertainty for the LDNS remains constant around $2.5 \pm 0.5$%, the energy uncertainty for LANSCE above 1 eV steadily increases up to $34 \pm 1$% for 37 keV. For the

same relative uncertainty at 10 keV, LANSCE thus requires a ToF distance of $13 \pm 1$ m, which reduces the neutron flux in comparison to the LDNS and a ToF distance of 1.8 m by a factor of $52 \pm 6$. Accordingly, ToF detectors can be operated at significantly shorter distances to the moderator at LDNS, which makes experimental setups more compact and neutron-efficient.

**Design of the collimation system and optimization of signal-to-background ratio.** A collimation system using multiple collimation layers with a converging recession was designed to maximize the number of neutrons at the detector while minimizing the presence of scattered neutrons close to the detector. Placing a collimator close to the moderator shields a large solid angle from neutron events and also reduces the presence of gamma events close to the detector. PHITS simulations were performed to optimize the geometry and the collimation system, see Fig. 3a, b. The detector shielding consists of 15-cm borated polyethylene (PE) to reflect fast neutrons coming from the source and additional 10-cm steel to reduce gamma events.

We investigated the neutron and background signal with PHITS simulations. Figure 3c displays the neutron and gamma flux as a function of time at the detector adjusted for the corresponding detector response. For arrival times earlier than 30 µs (19 eV), the gamma contribution is larger than the neutron count rate. For longer times, the neutron signal becomes dominant. The neutron and gamma signal are comparable in the important energy range of 1–19 eV (30–130 µs) in this configuration.

To assess the background from scattered neutrons, we simulated three different energy ranges. Figure 3d shows the neutron traces for <1 eV (orange), 1–10 eV (green), >10 eV (magenta). Most neutrons with energies below 1 eV (orange)

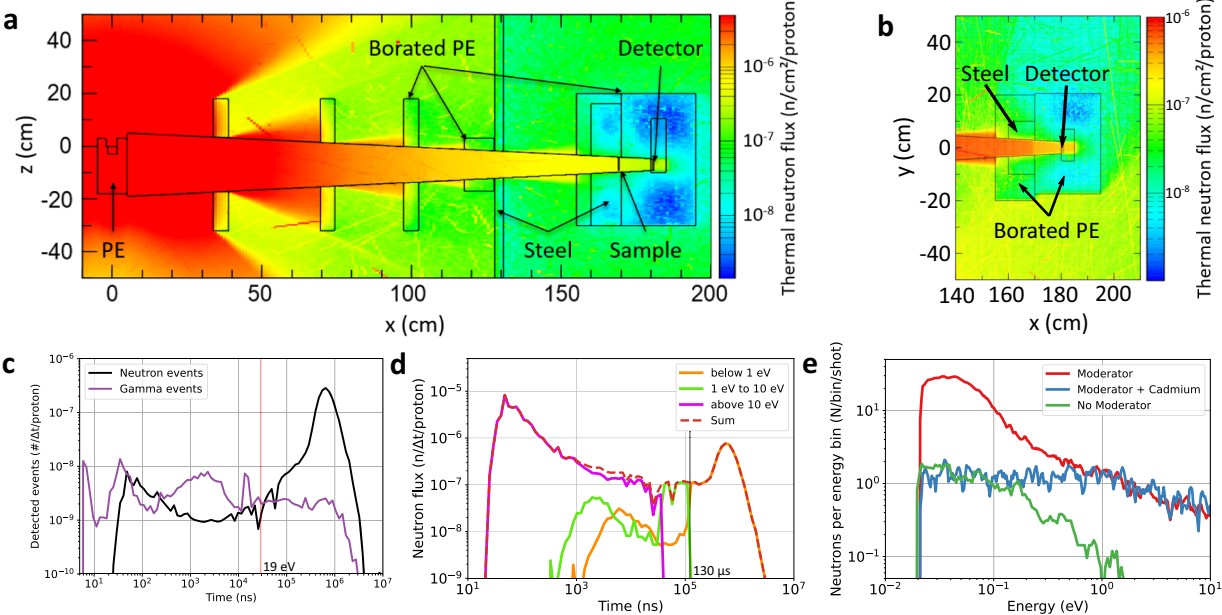

**Fig. 3 Optimization of the collimation setup and signal-to-background. a** PHITS simulation of the thermal neutron flux for the collimation setup (horizontal plane). The converging recess in the collimator parts results in a low background of scattered neutrons at the detector. **b** PHITS simulation of the thermal neutron flux of the detector shielding (vertical plane). A collimated epithermal neutron beam is produced at the detector position with low background. **c** Simulated arrival time of neutrons and gamma photons at the detector position including detector response. **d** Simulated arrival time of neutrons at the detector for different energy ranges (magenta: > 10 eV, green: 1–10 eV, orange: <1 eV). The time traces of the 1–10 eV and < 1 eV neutrons show signals before the corresponding moderator-detector flight time of 41 μs and 130 μs, respectively, which means that these neutrons do not have their origin in the moderator but in the borated-PE shielding. However, this neutron background is approximately one order of magnitude lower than the neutron signal from the moderator. **e** Measurement of the thermal neutron spectrum of the LDNS using the borated MCP as a detector. The red curve shows the measured neutron spectrum with a thermalized neutron peak at 25 meV. To assess the background for thermal energies, a 1-mm cadmium plate is used to block thermal neutrons (blue curve). Background for epithermal energies can be determined when the moderator is removed (green curve).

arrive at the detector after more than 130 μs, which corresponds to the ToF path of the moderator to the detector. However, the simulations reveal that 6.6% of the spectral neutron flux below 1 eV arrive earlier at the detector. These neutrons are moderated in the borated-PE shielding close to the detector and exhibit a contribution to the background of the ToF trace below 130 μs. A similar effect is found for the neutron flux at 1–10 eV.

The signal-to-background was investigated in the experiment for thermal energies using the borated MCP. Figure 3e (red) shows the measured neutron spectrum after a 2.7-mm tungsten plate with a thermalized neutron peak at 25 meV. To assess the background, a 1-mm cadmium plate is additionally placed in the beam path to block thermal neutrons (blue curve). The signal-to-background ratio in the thermal energy range is $(30 \pm 1)$:1. Above 1 eV, a signal-to-background of $(16 \pm 2)$:1 is estimated by comparing neutron spectra with (red curve) and without moderator (green curve).

**Neutron resonance spectroscopy**. The epithermal LDNS was used to perform a NRS measurement with a 2.7-mm-thin tungsten sample. Figure 4a (red) shows the transmitted neutron spectrum behind the sample adjusted for the detector response. At resonant energies, strong neutron absorption occurs leading to sharp dips in the neutron spectrum, which can be assigned to isotopes in the sample. In this measurement, several dips are observed at 4.3, 7.2, 10.2, and 14.0 eV. The first two match to the position of the $^{182}$W resonance at 4.15 eV and the $^{183}$W resonance at 7.6 eV[2]. The third and fourth dip hint at the presence of a $^{181}$Ta impurity in the sample with resonances at 10.3 eV and

14.0 eV and an additional resonance at 4.28 eV. The presence of the tantalum impurity was investigated with x-ray diffraction and a concentration of $(1.23 \pm 0.07)$% of the tantalum impurity was found (supplementary information file). The presence of tantalum was further confirmed by laser-driven neutron activation analysis (supplementary data file). The presence of the tungsten isotope $^{186}$W is in agreement with the data at 19 eV where the isotope has a wide resonance. At these energies, however, the experimental data is interrupted by several background events. A PHITS simulation has been performed to recreate the measured neutron spectrum using the experimental setup and a sample of natural tungsten and tantalum, see blue curve in Fig. 4a. The simulation agrees well with the experimental data. In particular, the position of the resonances in the 1–10 eV range shows good agreement with the predictions of the simulation. All resonances except for $^{186}$W show a larger than two sigma deviation from the statistical fluctuations observed at non-resonant energy ranges, see Fig. 4b. Due to the decreasing detection efficiency for higher energies and the large resonance width at 19 eV, the detected neutron flux is reduced close to the background event level as discussed above. Figure 4c shows the measured neutron spectrum at the $^{182}$W (4.15 eV) and $^{181}$Ta (4.28 eV) resonance with an increased energy binning width.

In the following, we evaluate the number of detected neutrons per energy bin and the resonance shape. Neutron resonances can be described by the Breit–Wigner function

$$N(E_\mathrm{n}) = -a \cdot \frac{\Gamma}{(E_\mathrm{N} - E_\mathrm{res})^2 + \Gamma^2/4} + \bar{N} \qquad (1)$$

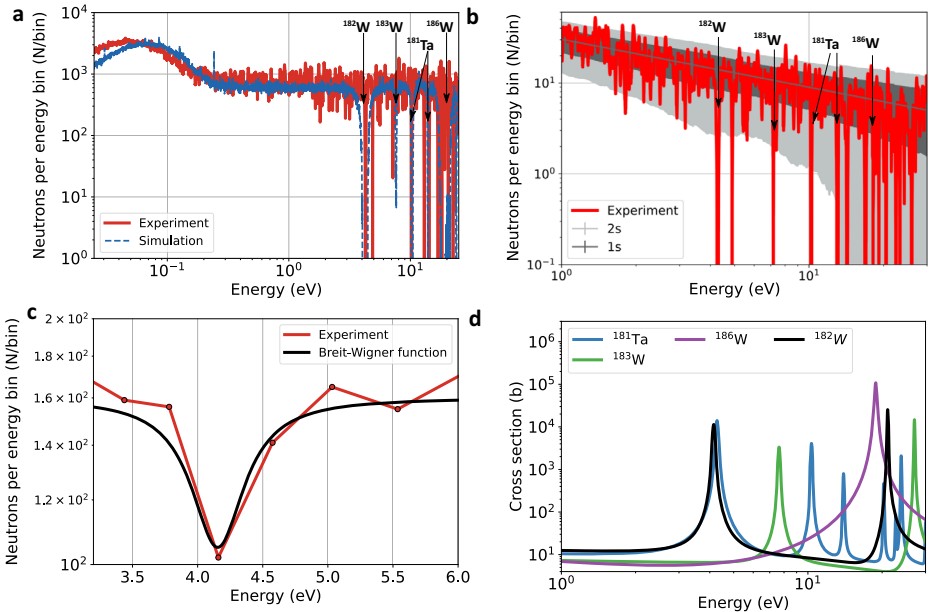

**Fig. 4 Neutron resonance spectroscopy with a LDNS. a** Measured neutron spectrum after transmission through a tungsten sample (red) and PHITS simulation (blue). Both in the experiment and the simulation, the isotopes $^{182}$W, $^{183}$W, $^{186}$W, and $^{181}$Ta can be identified. **b** The measured NRS signal is plotted (red) with the moving statistical fluctuation from the fit function $30 \cdot E_n^{-0.53}$ to assess the statistical significance of the isotope identification. The dark area corresponds to $1\sigma$ and the light gray area to $2\sigma$ deviation. **c** Measured neutron flux at the $^{182}$W resonance (red) with an increased bin width. The Breit–Wigner function (black) of the resonance of $^{182}$W with the position of 4.15 eV and the FWHM width $\Gamma = 0.55$ eV is in good agreement with the measurement. **d** All relevant resonances for tungsten and tantalum in the range of 1–25 eV using ref. [2].

with the energy width $\Gamma$ (FWHM), the resonance position $E_{res}$, the average number of non-resonant neutrons per bin $\bar{N}$, and a setup-dependent parameter $a$[34]. The black line in Fig. 4c represents the Breit–Wigner function for the $^{182}$W resonance with $\Gamma = 0.55$ eV, $E_{res} = 4.15$ eV, and $\bar{N} = 160$, calculated from the tabulated value for natural tungsten[2]. The measured spectrum shows good agreement with the expected resonance in shape and position. The central resonance energy was determined with a decreased bin width to 4.28 eV, which is within the calculated energy uncertainty of ±0.16 eV for this measurement.

The low number of 18 accumulated laser shots for this measurement requires a statistical analysis of the significance of the resonance data. For determining the resonance position, an average bin width of $E_n/85$ was used to have a sufficient energy resolution. The average number of neutrons per energy bin in the epithermal regime in this measurement was given via $N(E) = 30 \cdot E_n^{-0.53}$ eV$^{-1}$, displayed as the blue fit function in Fig. 4b). This corresponds to an average of $N = 14$ for the $^{182}$W resonance at 4.15 eV and to $N = 6$ for the $^{186}$W at 19 eV. Due to these low neutron numbers per bin, it has to be evaluated if these dips can also be the result of statistical fluctuations. The dark gray area in Fig. 4b visualizes the moving standard deviation of the measured signal from the fit. The light gray area visualizes the $2 \cdot \sigma$ confidence interval. It can be seen that all dips show a larger deviation than $2 \cdot \sigma$ from the average flux until 14 eV. This shows that all dips with an energy below 14 eV have a likelihood of above 97.5% to be caused by the presence of resonances. For higher energies, dips can also be caused by statistical fluctuations.

**Radiography using thermal neutrons**. In parallel to the NRS measurements, a radiography measurement with thermal neutrons was performed. Figure 5a and b shows the experimental setup as a side and front view. The sample setup is displayed in Fig. 5b and consisted of a 1-mm thin cadmium plate and an

indium sample with varying thickness ranging from 0.6 to 1.5 mm. The sample was measured through a 2-mm thin plate of lead. In addition, a personal neutron dosimeter (PND) was placed in the beam path below the cadmium sample. The borated MCP was gated to measure the attenuation of neutrons between 2 eV and 10 meV. Figure 5c shows the attenuation of the neutron flux in the radiography measurement with a single shot exposure. Using three laser shots, a homogeneous attenuation of $80 \pm 5\%$ behind cadmium and $32 \pm 8\%$ behind indium is observed, see Fig. 5d.

The attenuation data $a$ are further used to calculate the sample thickness distribution $d$ with $d = -\ln(1 - a)/\bar{\Sigma}$. This processing is possible when the material is known and $\bar{\Sigma}$ is determined. The retrieved thickness is shown in Fig. 5e and an average thickness of $0.83^{+0.06}_{-0.13}$ mm is obtained. This value deviates from the actual homogeneous sample thickness of 1 mm by 200 μm. The position of the cadmium sample could be determined with an accuracy of $2 \pm 1$ mm. The thickness distribution of the indium sample is shown in Fig. 5f and ranges from 0.4 to 1.6 mm, which is in good agreement with the measured sample thickness with an inhomogeneous thickness ranging from 0.6 to 1.5 mm.

**A simplified variant of NRI using thermal neutrons**. The capability of spatially detecting and identifying certain isotopes was further demonstrated with a simplified variant of NRI. A sample, behind a 2-mm lead plate, consisting of 2-mm cadmium (Cd) and 1.2-mm indium (In) sheets is wrapped in aluminum foil mimicking an unknown sample under investigation, see Fig. 6d. The borated MCP was read out in event mode so that all events are recorded with time stamps. The image is formed by using the data for different time intervals, which correspond to neutron energy windows. Figure 6a–c shows the retrieved detector images for a $E_n > 0.1$ eV, b 0.05 eV $< E_n < 0.1$, c $< 0.05$ eV. For $E_n > 0.1$ eV, the upper part has an average of $2.6 \pm 0.1$ n/cm$^2$ and the lower

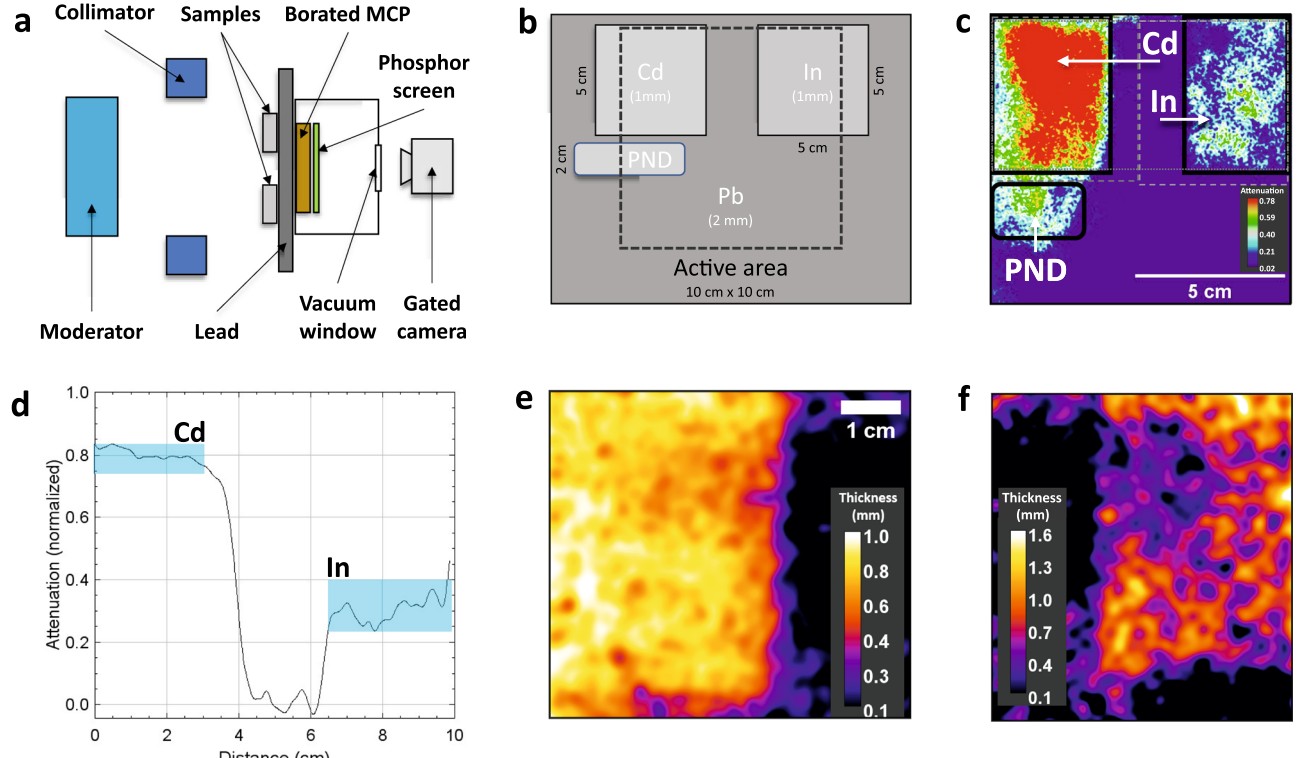

**Fig. 5 Thermal neutron radiography using an LDNS. a** Schematic setup of the thermal radiography measurement (top view). **b** Sample distribution across the detector (front view). One millimeter cadmium and indium sheets are placed on a 2-mm lead plate. **c** Thermal neutron radiography of the sample. The neutron attenuation is normalized to total absorption. The dotted line marks the area analyzed in (**d**) and the dashed lines marks the areas investigated in (**e**, **f**). The position and the shape of the In and Cd sample is visible. **d** Line-out through the Cd and In sample with the measured neutron attenuation normalized to total absorption. Cadmium shows 80 ± 5% absorption and indium 32 ± 8%. **e**, **f** Calculated thickness of the cadmium (**e**) and the indium (**f**) sheet using the measured attenuation and the known cross-sections.

part has $2.5 \pm 0.1 \, \mathrm{n/cm^2}$ and the Cd and In sheets cannot be resolved. This is in agreement to the PHITS simulation shown in Fig. 6e. When the energy window is shifted to lower energies between 0.05 eV and 0.1 eV changes the distribution and a stronger attenuation behind the cadmium can be seen with average fluxes of $6.55 \pm 0.05 \, \mathrm{n/cm^2}$ behind Cd and $9.2 \pm 0.1 \, \mathrm{n/cm^2}$ behind In. For energies below 0.05 eV, this difference increases to $17.6 \pm 0.3 \, \mathrm{n/cm^2}$ behind Cd and $52 \pm 1 \, \mathrm{n/cm^2}$ behind In. This agrees well with the simulation displayed in Fig. 6f. The large attenuation caused by the Cd resonance makes it possible to identify the presence and distribution of $^{113}$Cd within the field of view. Even though $^{113}$Cd has a resonance at low energies compared to other elements, this can still be seen as a simplified variant of NRI. The same measurement principle can be applied to epithermal energies when an efficient detector with the temporal and spatial resolution is used. Neutron resonances in the epithermal range for some isotopes of interest and the required measurement times, for a comparable 10-Hz laser system, are shown in Fig. 7.

## Discussion

We presented an optimized LDNS and demonstrated NRS in the epithermal range as well as neutron radiography and a variant of NRI in the thermal range, which were hitherto limited to large-scale neutron sources. An optimized converter and moderator design resulted in an epithermal neutron yield of $(4 \pm 1) \cdot 10^7$ neutrons per steradiant and laser pulse.

A comparison of the our LDNS with previous work shows the importance of moderator optimization. Mirfayzi et al.[14,17]

recently reported a fast neutron flux of $1.7 \cdot 10^9 \, \mathrm{n/sr/shot}$, which is similar to our LDNS, and a thermal and epithermal flux of $3 \cdot 10^6 \, \mathrm{n/sr/shot}$, which corresponds to a moderation efficiency of 0.2%. With the optimized moderator, we achieved a moderation efficiency of 2.5% with an average fast flux of $(1.6 \pm 0.4) \cdot 10^9 \, \mathrm{n/sr/}$ shot and a thermal and epithermal flux of $(4 \pm 1) \cdot 10^7 \, \mathrm{n/sr/shot}$. The optimized thermal and epithermal flux enabled the demonstration of laser-driven NRS, thermal neutron radiography, and a simplified variant of NRI.

With the NRS setup, neutron resonances up to an energy of 14 eV are identified using 18 laser shots. As the efficiency of our detectors decreases for higher energies, more shots or more efficient detectors are required to resolve resonances at higher energies. The relatively low statistics of our measurements is the reason why we cannot estimate the concentration of each isotope. However, the obtained results can be used to predict the number of laser shots needed to determine the presence of a given resonance in a sample. For an isotope to be detected with this technique, the attenuation of the neutron flux by the resonance has to be larger than the statistical fluctuation of the neutron signal at the same energy. On the one hand, the number of measured neutrons is limited per non-resonant energy bin $N$. Due to the Poissonian nature of the measurement process, we assume that the relative fluctuation for this number is given via

$$\sigma_{\mathrm{rel}} = \frac{1}{\sqrt{N}} \qquad (2)$$

On the other hand, the detectability is limited by the attenuation

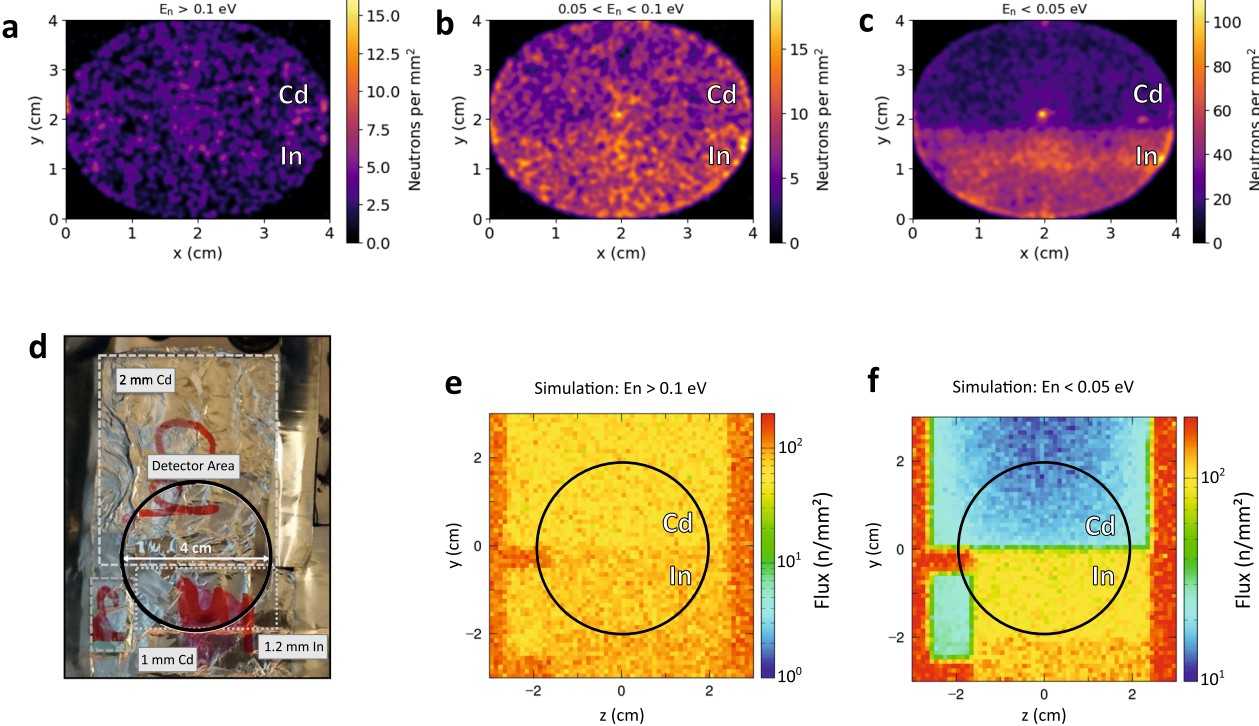

**Fig. 6 Neutron resonance imaging (NRI) with a LDNS. a–c** Spatial distribution of detected neutron events for the NRI measurement. **a** The MCP detector signal shows all non-resonant neutron events with arrival times corresponding to energies > 0.1 eV. **b** The MCP detector signal shows events attributed to neutron energies between 0.05 and 0.1 eV. An increased absorption behind the 2-mm cadmium sheet is visible. **c** shows the distribution of neutron events with energies below 0.05 eV. A strong attenuation is visible behind the cadmium sheet and a small attenuation is observed behind the indium sample. **d** Sample distribution in front of the borated MCP detector. The active detector area has a diameter of 4 cm (black circle). The upper half is covered with 2-mm thick Cd and the lower half with 1.2-mm thick In (dashed white rectangles). **e**, **f** PHITS simulations for (**a**, **c**). The simulation (**e**) predicts 76 ± 5 n/cm$^2$ behind Cd and 94 ± 4 n/cm$^2$ behind In. **f** predicts 16 ± 1 n/cm$^2$ behind Cd and 111 ± 6 n/cm$^2$ behind In.

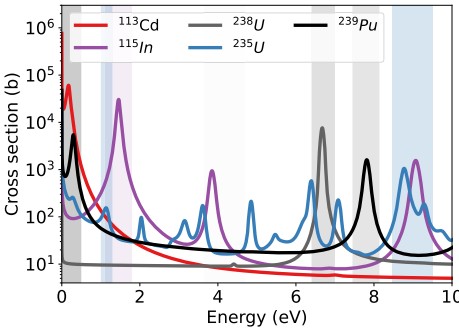

**Fig. 7 Distribution of neutron resonances from Table 1.** The shades areas indicate the corresponding energy ranges. Cross-sections taken from ref. [2].

caused at resonant energies, which is calculated by

$$\frac{I}{I_0} = \exp\left(-\hat{\Sigma} \cdot d \cdot f_{\mathbf{r}}\right) \quad (3)$$

with the macroscopic cross-section $\hat{\Sigma}$ of the resonance, the sample thickness d, and the fraction of the isotope $f_{\mathbf{r}}$ inside the sample. Assuming that the attenuation has to be larger than $2\sigma_{\text{rel}}$ of the fluctuations this leads to a minimal amount of neutrons per bin that is necessary to observe a resonance:

$$N > \left(\frac{2}{1 - \exp\left(-\hat{\Sigma} \cdot d \cdot f_{\mathbf{r}}\right)}\right)^2 \quad (4)$$

Here, $N$ depends on the transmissivity of the sample for non-resonant energies. To calculate the necessary number of neutrons

per bin without a sample $\hat{N}$, Eq. (4) has to be adjusted to

$$\hat{N} > \left(\frac{2}{1 - \exp\left(-\hat{\Sigma} \cdot d \cdot f_{\mathbf{r}}\right)}\right)^2 \cdot \exp\left(\sum_{i=1}^{k} \Sigma_i \cdot d_i \cdot f_i\right) \quad (5)$$

with the macroscopic cross-section $\Sigma_i$, the thickness $d_i$, and the fraction $f_i$ of all other present isotopes in the sample.

We can now perform calculations of samples, which consist of nuclear material and can have a high impact, for example, in the investigation of nuclear fuel or illicit material trafficking. We assume the presence of $^{235}$U in a 1-cm thick sheet of natural uranium as an example. With Eq. (5), the required number of neutrons has to be larger than 78 neutrons per bin at the 8.75 eV resonance of $^{235}$U . A comparison of this value with the NRS of the tungsten impurity at approximately the same energy with 9.5 N/bin after 18 shots and transmissivity of 83%, a detection of 0.72% $^{235}$U in 1-cm natural uranium would require 123 laser shots with the presented setup or a source emission of $2.5 \cdot 10^{12}$ fast neutrons. Modern laser systems are developed with a repetition rate of 1–10 Hz[9,35,36] and can achieve proton cut-off energies of ~60 MeV[37], which is comparable to the present work. Such PW-class laser systems will enable laser-driven proton and deuteron acceleration from replenishing jet targets[38–40] that operate at repetition rates of Hz. Using our experimental simulation results, an optimized LDNS with a repetition rate of 1 Hz in the epithermal range would facilitate a measurement of a $^{235}$U impurity in the configuration described above within two minutes. For lasers in the 200 TW range that operate at repetition rates of 10 Hz, our simulations predict a reduction of the neutron yield by a factor of 2.5[29] due to the lower proton/deuteron cut-off energy of 20 MeV[39]. The measurement time with such a 10 Hz

**Table 1 Neutron resonances of interest.**

| Isotope | Energy (eV) | Number of shots | Time at 10 Hz |
|---------|-------------|-----------------|---------------|
| [113]Cd | 0.001–0.05 | 17 | 1.7 s |
| [239]Pu | 0.001–0.50 | 10 | 1 s |
| [115]In | 1.10–1.80 | 1.7E3 | 0.5 h |
| [235]U | 1.00–1.30 | 3.1E3 | 0.9 h |
| [182]W | 3.65–4.70 | 6.0E3 | 1.7 h |
| [235]U | 8.45–9.50 | 1.7E4 | 4.8 h |
| [238]U | 6.40–7.00 | 2.3E4 | 6.4 h |
| [239]Pu | 7.45–8.14 | 2.6E4 | 7.2 h |

The number of shots needed for an NRI measurement to resolve isotopes with a given area density of $1 \cdot 10^{21}$ atoms/cm$^2$. The last column indicates the time needed for this measurement if a comparable laser system is operated at 10 Hz.

laser proton accelerator facility would reduce would be increased to about 5 min.

With the demonstration of a variant of NRI, we have shown that it is possible to identify a cadmium sample next to an indium sample even though those two materials have similar atomic densities but largely different resonance structures. Also in this case, the NRI measurement can be strongly improved with a higher neutron flux, i.e., by using more laser pulses. In addition, the measurement can be improved by a normalization to the response of the pixelated detector with a homogeneous neutron beam, a so-called flat-field measurement. This could not be performed in the NRI experiment due to time constraints.

Our results can be used for this application scenario to extrapolate the required number of shots, which are required for an NRI measurement of a sample consisting of different isotopes. We, therefore, integrate the number of neutrons at a given resonance and compare it to the NRI measurement taking into account the detector response function and the changes in the moderated neutron spectrum. We then calculate the number of shots which are needed to achieve a NRI measurement for an isotope with the same area density of $1 \cdot 10^{21}$ atoms/cm$^2$ as for [113]Cd in the NRI measurement. The projection for different isotopes is presented in Table 1. For an energy window of 1 meV to 0.5 eV, it would require ten shots to identify a [239]Pu sample with this technique, as [239]Pu has a similar low energy resonance as [113]Cd at 300 meV. For a laser system firing at a repetition rate of 10 Hz, this measurement would be recorded in one second. In order to analyze more complex samples using NRI at a LDNS, the collimation system needs further optimization to increase the signal-to-noise ratio. Furthermore, the influence of the gamma flash on the MCP must be reduced by either blocking or obscuring the line of sight from the MCP to the laser-target interaction. This should result in an increased sensitivity for epithermal neutrons, as the MCP is not saturated by the gamma radiation. With such a setup, an unknown sample could be investigated in the epithermal energy range by measuring the spatial absorption for different neutron energies.

A LDNS, which utilizes our optimized setup and has a similar proton/deuteron yield with a repetition rate of 10 Hz, would produce a thermal and epithermal neutron flux of the order of $10^4$ n/cm$^2$/s at the detector. This is indeed three orders of magnitude lower than LANSCE but it is not far from the thermal flux of the GELINA facility providing $5.4 \cdot 10^5$ n/cm$^2$/s[41]. Scaling the results of this work to present laser technology at 10 Hz would enable thermal and epithermal LDNS with $10^7$ and $10^8$ neutrons/sr/s[29,42,43].

However, optimized LDNS may find applications of high impact as they are relatively compact and the ToF distance can be in the range of a few meters due to the short ion pulse duration.

Compact LDNS thus open up the capability of neutron-based material inspection at small laboratories or industrial sites. With the detection of isotopes such as [235]U in natural uranium or [239]Pu, LNDS can be applied to prevent illicit nuclear materials trafficking and proliferation. LDNS may further play an important role in nuclear safeguard[44] and in the inspection of used nuclear fuel rods. In conclusion, optimized LDNS exploiting PW-class lasers with repetition rates of 1–10 Hz[36] will open prospects for the emerging field of nuclear photonics and its applications.

## Methods

**Laser-driven neutron source.** The results were obtained in the user experiments P153 and P197 at the PHELIX laser at the GSI Helmholtz Centre for Heavy Ion Reseach[22]. The laser was operated at $103 \pm 9$ J with a pulse duration of $0.6 \pm 0.1$ ps and a focal spot with $4.0 \pm 0.5$ μm diameter (FWHM). The temporal contrast of laser peak power to amplified spontaneous emission was measured to $10^{-12}$ using a third-order cross-correlation system. The incidence angle of the laser to the target normal was set to 6.5°. Deuterated polystyrene foils were used as targets with a thickness in the range of 400–1100 nm. The mean proton cut-off energy was measured with a stack of radiochromic films to 37 MeV with a standard deviation of 13 MeV. The mean cut-off energy of deuteron ions and the proton-to-deuteron ratio was measured with a Thomson parabola spectrometer for some shots in advance of the neutron production experiment. As a converter for neutron production, three different designs were used. First, a 3-cm long beryllium cylinder with a diameter of 3 cm protected by a 1-mm thick polyamide ablation shield was used. In the second design, the polyamide was replaced by 2-mm-thick lithium fluoride. The third design was a pure lithium fluoride cylinder with a 38 mm diameter and a 25 mm length. All converters, except LiF, were placed inside a cylindrical copper casing with a wall thickness of 2 mm. The laser-facing side of the casing remained open. The fast neutron yield was measured with neutron dosimeter bubble detectors[30]. The average number of fast neutrons produced during this beamtime was $(1.6 \pm 0.4) \cdot 10^{10}$ neutrons per shot. All neutron detectors were placed outside the vacuum chamber. For each detector, a 1-mm thin aluminum window in the target chamber was used. The fast neutron spectrum was measured with a NE102 organic scintillator coupled to a Hamamatsu R1828-01 photomultiplier. The response was adjusted for the attenuation by a 20-cm lead shielding, which was used to reduce the gamma signal. In addition, the energy-dependent light production of neutrons was accounted for by calculating the deposited energy in the scintillator with Monte Carlo simulations. The relation between deposited energy and light output in organic scintillators was taken from O'Rielly et al.[45].

**Neutron resonance spectroscopy measurement.** For the NRS measurement, 18 shots were accumulated with an average source emission of $(2 \pm 1) \cdot 10^{10}$ fast neutrons per shot. In this setup, the neutron resonance transmission method was used[5]. Neutrons were collimated by five borated polyethylene sheaths with a thickness of 5 cm each and a decreasing recess towards the detector outside the target chamber. The borated MCP detector was placed at a distance of $180 \pm 1$ cm perpendicular to the moderator side surface. The 2.7-mm thick sample was placed at a distance of 10 cm in front of the detector to avoid the detection of prompt absorption gammas produced inside the sample. The shielding in front of the NRS detector consists of 10 cm of steel and 15 cm of borated polyethylene between the detector and the source to reduce the scattered neutron background. Single neutron events were detected with a borated MCP with 40 mm diameter, which has a sufficiently high temporal resolution. The readout of the MCP was performed in single event mode with a time resolution of 25 ns. The decreasing cross-sections of [10]B for higher neutron energies result in a lower sensitivity, which was corrected for by the detector response function, using the known (n,α) cross-section of [10]B[46]. Each neutron event was assigned to energy via time-of-flight. The energy binning in the processing algorithm for Fig. 4a changes for each bin as a function of their energy $\Delta E_n = E_n/B$. The binning resolution $B$ was varied from 70 to 100 for each measurement and all data points were subsequently smoothed to reduce the influence of the binning on the resonance shape and to increase the energy resolution. For Fig. 4b, no smoothing was applied and $B$ was set to 10 to decrease statistical fluctuations for a more accurate measurement of the resonance width. The minimal detectable neutron energy was limited to 20 meV due to the maximum record length of the read-out electronics. The MCP was encapsulated in a Faraday cage to shield it against electromagnetic pulses from the laser-target interaction.

**Monte Carlo simulations.** All simulations were performed using the Monte Carlo code PHITS[26] (version 3.16). In Fig. 4, a proton spectrum by TNSA with a cut-off energy of 54 MeV was used to generate fast neutrons in a beryllium target by (p,n) reactions. These fast neutrons were slowed down in the moderator made from polyethylene, as shown in Fig. 2d. In the simulations, the proton reactions were modeled using version 4.6 of the Intra-Nuclear Cascade of Liège (INCL) model[47]. The JENDL-4.0[48] nuclear data library was used to calculate the neutron transport for neutrons below 20 MeV. For the conversion efficiency from protons to

neutrons, the FENDL library version 3.1d[49] was used. The tungsten sample in the simulation was assumed to be of 50% natural composition and 50% tantalum in order to increase the visibility of tantalum resonances. The target chamber was modeled as plain steel, the floor as regular concrete, and the collimation system as 5% borated polyethylene.

**Thermal neutron radiography measurement**. The thermal neutron radiography measurement was performed using the Neutronic [i] detector system (Photonis BV, Netherlands) combined with an intensified gated camera (ProxiKit from ProxiVision GmbH, Germany). Neutronic [i] is a $10 \times 10$ cm$^2$ borated MCP coupled to a fast phosphor screen inside a vacuum housing. The phosphor screen was imaged from the rear side through a window using the ProxiKit camera. The entire setup was shielded against optical light as well as scattered neutrons by 10 cm of borated polyethylene. The thermal neutron radiography in Fig. 5b was recorded with a single shot with 124 J laser energy on target with $(6 \pm 3) \cdot 10^8$ fast neutrons produced. The image was taken 59 μs after the laser impact with an exposure time of 757 μs. For the moderator-detector distance of 1.14 m, this corresponds to 10 meV to 2 eV. The attenuation is calculated via a background subtraction and a normalization explained in detail in ref. [29]. For Fig. 5d–f, three shots were accumulated and averaged to reduce statistical fluctuations. For the thickness calculation, the attenuation of the neutron spectrum was simulated after a reference indium and cadmium sample. The spectrum was adjusted for the detector response. The data were then used to define an average macroscopic cross-section $\bar{\Sigma}$, for which the thickness $d$ was calculated from the attenuation $a$ according to $d = -\ln(1-a)/\bar{\Sigma}$.

**Neutron resonance imaging**. The NRI detector was placed at a distance of 1.45 m from the moderator. The detector was a borated MCP (NeuView MCP provided by Surface Concept GmbH) with 40 mm diameter and a high temporal resolution, which allow the detection of single neutron events by the means of a delay line anode. The pixel size of the detector was $36 \times 48$ μm$^2$ with a temporal resolution better than 240 ps for the MCP itself. The borated MCP plate of the detector measured thermal and epithermal neutrons up to ~1 eV. Above roughly 200 meV to 250 meV, the detector exhibits reduced sensitivity to neutron events, which is equivalent to arrival times earlier than 210 μs after the shot. This is most likely due to saturation of the MCP owing to the gamma radiation, which is produced by the laser-target interaction. Due to the thickness of the MCP and the speed of the thermal neutrons, the temporal resolution for 25.3 meV neutrons is around 380 ns. The Cd sheet and the In layer had a thickness of 2 mm and 1.2 mm, respectively. The detector was shielded by an additional 2 mm plate of lead against X-ray background in the beam direction. In all other directions, the detector was shielded by 10 cm of borated polyethylene. The bright spot in the center of the detector is a known artifact of the detector. In total, 17 shots were accumulated for the NRI measurement in order to minimize statistical fluctuations. The MCP trigger was set to 100 μs after the laser shot and recorded neutron events for a total of 3 ms.

## Data availability

Supplementary Information is available in the online version of the paper. Datasets analyzed during this study are provided as a source data file. Source data are provided with this paper.

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

## Acknowledgements

The results are based on the experiments P197 and P153, which were performed at the PHELIX facility at the GSI Helmholtzzentrum für Schwerionenforschung, Darmstadt (Germany) in the frame of FAIR Phase-0. We want to thank the PHELIX operator team for their excellent user support. We thank the group of Prof. Bjoern Winkler for the measurement of the lattice parameter of the tungsten sample. C.R., T.E., and M.Z. are supported by the Hessian Ministry for Science and the Arts (HMWK) through the LOEWE Research Cluster Nuclear Photonics. S.S. acknowledges support from Trumpf GmbH & Co. KG.

## Author contributions

M.Z. developed the concept, optimized the setup, and conceived the study. M.Z., S.S., A.K. N.M., S.Z., and M.H. performed the experiment. M.Z. and S.S. analyzed the results. M.Z. and S.S. performed the simulations. A.T., T.A., G.S., and T.E. provided the targets. S.D.P., O.M., R.J.A., T.K., and M.P. provided detectors. S.C.V. helped with the detector calibration. M.Z., C.R., S.S., and M.R. wrote the manuscript. V.B. supervised the laser operation. M.R. supervised the project. All authors commented on the manuscript.

## Funding

## Competing interests

The authors declare no competing interests.
