## [Peer Review File · Nature Communications]

Demonstration of non-destructive and isotope-sensitive material analysis using a short-pulsed laser-driven epithermal neutron sourceReviewers' comments:

Reviewer #1 (Remarks to the Author):

Referee report "Demonstration of non-destructive and isotope-sensitive material analysis 2 using a laser-driven epi-thermal neutron source" by Zimmer et al.

I recommend publication in Nature Communications.

In this manuscript Zimmer et al. report on the first production of epi-thermal neutrons with a laser-driven neutron source and on the first proof-of-principle demonstration of the use of such a source for Neutron resonance spectroscopy, neutron radiography, and neutron resonance imaging.

The paper is well written, understandable, and technically sound. It suffers from a dearth of statistics given by the nature of the experimental facility, which only allows for a limited number of shots in an experimental campaign. The authors should address this and discuss reproducibility and confidence intervals of their results, this however does not invalidate the presented findings. While the list of references can always be more exhaustive, the authors cite the relevant work in the field.

While the underlying physics of neutron production and moderation is of course not new, the methods being applied routinely at large accelerator or reactor-based neutron sources, their successful application with a laser-driven source is a first. It is of considerable importance not only to the physics community but also to material science as well as a large field of applications. Multiple papers in the last few years have shown progress in demonstrating laser-driven neutron sources using either laser-accelerated ions and electrons/gamma-rays, and these papers have already speculated on their possible use. Yet they have not yet widely influenced the larger community of neutron source users and potential hosts, to take laser neutron sources into account when planning future work or facilities. Only MeV neutrons had been produced so far, and while it is evident for a nuclear physicist that these can of course be moderated, it was one step too far removed from actual applications. The question that remained unanswered was whether the laser-produced neutrons could be moderated in an efficient way that would allow actual applications. This manuscript answers this question and demonstrates that this can indeed be done. With the advent of more efficient, higher repetition rate PW class lasers, as installed e.g. in the three European ELI-facilities, neutron sources now become accessible to a much wider range of institutions. As lasers shrink and become even more compact and cost effective, this trend will only accelerate. It is my opinion that this manuscript is of considerable interest to a wider community and deserves publication in Nature Communications.

Bjorn M.

Hegelich UT

Austin

Reviewer #2 (Remarks to the Author):

The manuscript by Zimmer et al. describes the demonstration of a laser-driven epithermal neutron source, which is then applied to precursory demonstrations of neutron resonance spectroscopy, neutron radiography, and neutron resonance imaging. The techniques described represent important steps toward achieving compact laser-driven neutron sources for high-impact applications such as non-destructive evaluation and isotope-specific material characterization.

In general, the manuscript was very difficult to follow, and attempted to cover too many techniques without adequate explanations or details on any. Partially, it is the format of Nature Communications, which makes use of a relatively brief article followed by methods and the more extensive supplementary material. However, the bulk of the article does not convey adequately the methodology or validity of the approach without extensive referencing to the methods or supplementary material, which makes it incredibly confusing for the reader. For this reason alone, this article is likely better suited for a different journal.

The article would then also be far more informative if it did not attempt to demonstrate three different applications (neutron spectroscopy, thermal neutron radiography, neutron resonance imaging) simultaneously, but rather investigate each of these separately and attempt to achieve higher quality data than what is shown herein. Each of these applications is very challenging and requires extensive development — rigorous R&D into any one of them with results that are validated, repeatable, and expandable to future experiments or uses would be very worthy and laudable. However, what is shown in this publication is likely as harmful to the translation of these techniques to application space as it is for pushing the field forward — as the results are not terribly convincing and discussion of the results lack rigorous error bars or consideration of how to improve the results and generalize the experimental setup such that others may use or build off of these techniques.

The novelty and significance of this work is not well articulated. There have been other one-off demonstrations of short-pulse laser-driven NRS, NRI, and neutron radiography. This paper would be strengthened by either discussing how this latest work shows an improvement on other early work, or more clearly call attention to the fact of the versatility of the laser-driven neutron source that these can be done simultaneously or with just minor adjustments to the source or experimental geometry.

Further detailed comments follow regarding the manuscript:

- In the introduction, it is claimed that the laser driven neutron sources could potentially compete with spallation or accelerator-based neutron sources such as LANSCE. However, few quantitative details are given on “the requirements on the neutron flux and temporal pulse width,” and whether these new laser driven neutron sources can yet meet those requirements (and if not, how far are they?). For each of the NRS, NRI, radiography applications, how do the results obtained here compare to the state-of-the-art, what advantages or disadvantages does the laser-driven approach provide, and how much more development is needed to make them viable?

- This reviewer is relatively senior and familiar with the scientific literature in the high-intensity laser science field. However, I have never before come across the notation using the parenthesis (e.g., 103(9) J, or peak intensity of $2.0(3) * 10^{20} \text{W/cm}^2$). Is the number in parentheses meant to denote an alternate case or an error bar or an additional significant digit?

- The concept and characterization of the laser-driven epithermal neutron source is notable and is alone worthy of a publication, if further details could be given on how the experimental parameters scaled or influenced the resulting spectrum, and exactly which spectrum was best for which application. For example, in the Methods, it is mentioned the range of deuterated polystyrene foil targets used, and that three different converter designs were tested. However, there is no further description of which design ended up being used for each of the demonstrations shown in the paper.

- Figure 1 (d) and (e) show the simulated moderator cross section in color bar units of $\text{n/cm}^2/\text{proton}$. But this is not explained at all in the paper — By the time the fast neutrons hit the moderator, the conversion from protons has already occurred in the converter foil. So what are (d) and (e) showing?? 1(b) appears to show an example of a fast neutron spectrum, but for what target and laser conditions, and/or if it is representative, it would be good to show error bars or a range of spectra achieved for this investigation. As this manuscript is primarily an experimental demonstration, specific geometries and setup are quite pertinent, so Fig. 1 of the supplementary information was very informative, and some combination of Fig. 1 in manuscript and Fig. 1 in supplementary would be helpful.

- MCP = micro channel plate or multi channel plate?

- What code was used for the NRS simulations? It is never explicitly said or details given about this simulation. Fig. 2(b) would be more convincing if the detector corrections and binning adjustments were applied to multiple resonances and could be shown to consistently match Breit-Wigner. This could also be extended to make a statement about the relative isotopic concentrations, and/or if the measurement requires improvement to extract this, it should be stated.

- Is Fig. 2(d) caption a typo? It appears that <1 eV neutrons should arrive later at the detector? Perhaps this needs more explanation.

- Fig. 3(a) is unintelligible. The reader has no idea what they are looking at, and there is no scale. The black boxes drawn differ from those in 3(b), so this only adds confusion. Without a scale or error bars, the reader cannot ascertain whether the retrieved thicknesses and positions using thermal neutron radiography are good or terrible.

- The paper could be improved wholly by citing error bars for all measurements. Discussion on how those error bars were derived can be placed in the Supplementary material.

- An honest evaluation of the thermal neutron radiography technique would include a thickness and uniformity retrieval of the In sample as well.

- Neutron Resonance Imaging is indeed a very powerful technique, however, it is not clear that the demonstration here is really a demonstration of NRI at all versus just neutron attenuation. The manuscript claims that the detected neutrons is influenced by the ^{113}Cd resonance, but there is no information given on what energy the resonance occurs at or how much attenuation is expected. A more convincing demonstration would have used ^{113}Cd covering a portion or the detector, and another material of a matching areal density (or material of very different resonance) to show differences in the attenuation.

Reviewer #3 (Remarks to the Author):

Neutrons are a valuable tool for non-destructive material investigation. So far, most neutron applications have been limited to large-scale facilities such as nuclear research reactors, spallation sources, and accelerator-driven neutron sources. With the development of the laser system, laser-driven proton accelerators enable compact neutron source. Use for the laser-driven neutron source (LDNS) have been so far limited to neutron radiography. The authors report the attempt to use a LDNS to neutron resonance spectroscopy (NRS) and the neutron resonance imaging (NRI).

The system is based on the laser which gives peak intensity of $2.0\text{E}20$ W/cm². Accelerated protons and deuterons are directed to a converter to generate neutrons via (p,n) and (d,n) reaction to yield neutrons. Neutrons from the converter enter the moderator made of polyethylene, which was optimized to give the maximum yield of 1~10 eV. The epi-thermal neutrons propagate through the sample, then detected afterward.

The NRS measurement was successfully demonstrated using a tungsten sample. The transmitted spectrum clearly shows the sharp dips corresponding to the three tungsten isotopes, as well as impurity isotope Ta. Also, signal-to-background ratio 30:1 was obtained. Concerning the performance for thermal neutron radiography, neutron transmission measurement through the cadmium and indium sample demonstrates that the system can be used to measure distribution of the sample thickness, and the spatial position of the sample was determined with an accuracy of 2 mm.

Then the authors attempted the NRI measurement. However, I think the measurement did not demonstrate the scope of NRI. As I understand, it is the approach to identify the material from the resonance position and to find their distribution (location) over the sample as well as its amount. The authors contrasted the image between the low-energy region (0.004 – 0.1 eV) and high energy image (0.1 – 1 eV) using cadmium neutron absorber. I admit that the strong absorption of Cd113 is associated with the resonance at the thermal energy region. But this is a very special case found in Cd113. In general, NRI should be adopted to more general material who has resonances at higher energies.

Actually, the present NRS experiment shows the resonances in the neutron energy range of 1 -10 eV, which is larger than the analysis given in FIG.4, by which the authors claim the demonstration of NRI. Thus, it is difficult to agree that the authors proved the applicability of the NRI using LDNS. I believe analysis of different material needs to be done to claim the success of NRI.

From these reasons, I do not recommend this article to be published in Nature Communications.

Reviewer #1:

“I recommend publication in Nature Communications. In this manuscript Zimmer et al. report on the first production of epi-thermal neutrons with a laser-driven neutron source and on the first proof-of-principle demonstration of the use of such a source for Neutron resonance spectroscopy, neutron radiography, and neutron resonance imaging.

The paper is well written, understandable, and technically sound. It suffers from a dearth of statistics given by the nature of the experimental facility, which only allows for a limited number of shots in an experimental campaign. The authors should address this and discuss reproducibility and confidence intervals of their results, this however does not invalidate the presented findings. While the list of references can always be more exhaustive, the authors cite the relevant work in the field.

While the underlying physics of neutron production and moderation is of course not new, the methods being applied routinely at large accelerator or reactor-based neutron sources, their successful application with a laser-driven source is a first. It is of considerable importance not only to the physics community but also to material science as well as a large field of applications. Multiple papers in the last few years have shown progress in demonstrating laser-driven neutron sources using either laser-accelerated ions and electrons/gamma-rays, and these papers have already speculated on their possible use. Yet they have not yet widely influenced the larger community of neutron source users and potential hosts, to take laser neutron sources into account when planning future work or facilities. Only MeV neutrons had been produced so far, and while it is evident for a nuclear physicist that these can of course be moderated, it was one step to far removed from actual applications. The question that remained unanswered was whether the laser-produced neutrons could be moderated in an efficient way that would allow actual applications. This manuscript answers this question and demonstrates that this can indeed be done. With the advent of more efficient, higher repetition rate PW class lasers, as installed e.g. in the three European ELI-facilities, neutron sources now become accessible to a much wider range of institutions. As lasers shrink and become even more compact and cost effective, this trend will only accelerate. It is my opinion that this manuscript is of considerable interest to a wider community and deserves publication in Nature Communications.”

We thank Referee #1 for his positive report and are pleased that he strongly supports the publication of our results in Nature Communications. We agree with the conclusion that this manuscript answers the heavily debated question if laser-driven neutron sources can be operated efficiently enough in the epi-thermal range for applications which were only proposed in previous publications. We also appreciate his constructive critique, which we address in the revised manuscript.

Points of main concern:

“It suffers from a dearth of statistics given by the nature of the experimental facility, which only allows for a limited number of shots in an experimental campaign. The authors should address this and discuss reproducibility and confidence intervals of their results, this however does not invalidate the presented findings.”

We agree that especially the NRS and NRI measurement is limited by the low statistics, which cannot be overcome at the PHELIIX laser. We have added an evaluation of the significance of the resonances based on an analysis of the statistical fluctuations, which defines the signal-to-noise ratio. We further added a comparison with simulations and evaluated the influence of background gamma events. In addition, we discuss the reproducibility and requirements for analyzing other resonances and clarified that the technique in its current form is limited to identify the presence of an isotope but not its concentration. We further defined a minimal number of laser shots required for detecting the presence of a given isotope based on resonance height and isotope concentration.

Reviewer #2:

“The manuscript by Zimmer et al. describes the demonstration of a laser-driven epithermal neutron source, which is then applied to precursory demonstrations of neutron resonance spectroscopy, neutron radiography, and neutron resonance imaging. The techniques described represent important steps toward achieving compact laser-driven neutron sources for high-impact applications such as non-destructive evaluation and isotope-specific material characterization.”

We thank Reviewer #2 for the careful reading of the manuscript and his/her constructive critique and the comprehensive suggestions for improvement. We are glad that he/she highlights the significance of the work and supports publication of the results by saying “the concept and characterization of the laser-driven epithermal neutron source is notable and is alone worthy of a publication”.

We have followed his/her advices and strongly revised the manuscript. We added a full characterization of the epi-thermal source and a description of the development of the source.

Points of concern:

“In general, the manuscript was very difficult to follow, and attempted to cover too many techniques without adequate explanations or details on any. Partially, it is the format of Nature Communications, which makes use of a relatively brief article followed by methods and the more extensive supplementary material. However, the bulk of the article does not convey adequately the methodology or validity of the approach without extensive referencing to the methods or supplementary material, which makes it incredibly confusing for the reader. For this reason alone, this article is likely better suited for a different journal.”

The development and optimization of a laser-driven neutron source is a major step in the field that can lead to one of the only real-world applications of laser-based proton accelerators as pointed out by referee 1. Based on the critique of referee 2, we completely revised the manuscript and describe the development and optimization of the laser-driven neutron source in detail.

“The article would then also be far more informative if it did not attempt to demonstrate three different applications (neutron spectroscopy, thermal neutron radiography, neutron resonance imaging) simultaneously, but rather investigate each of these separately and attempt to achieve

higher quality data than what is shown herein. Each of these applications is very challenging and requires extensive development; rigorous R&D into any one of them with results that are validated, repeatable, and expandable to future experiments or uses would be very worthy and laudable.

We appreciate the comment of the referee and agree that demonstrating applications of a laser-driven neutron is very challenging. However, the techniques have been demonstrated at accelerator-based neutron sources such as GELINA. The focus of our paper is the transfer of these techniques to a laser-driven neutron source, which was claimed in previous applications but has not been shown yet. We strongly revised the manuscript in order to explain, which optimization was required to facilitate the demonstration of the techniques. For example, the measurements were enabled by optimizing the moderation efficiency by a factor of 10 compared to previous works using state-of-the-art modelling. We think that the presented work will be used as a reference for the development of laser-driven neutron sources.

The latest progress in laser proton acceleration is very encouraging for laser-driven neutron sources. For example, Ziegler et al. (Scientific Reports 7338, 2021) recently demonstrated that 60 MeV class proton beams can be accelerated with a PW-class laser. Gauthier et al. (Applied Physics Letters 111, 2017) have shown that cryogenic jets can be used as targets for laser proton acceleration. Together with the development of PW-class lasers with repetition rates of 1-10 Hz, e.g. the HAPLS laser at ELI Prague (Sistrunk et al., CLEO 2017), laser-driven neutron sources can be developed with high application potential.

Our experimental results are supported by state-of-the-art modeling of neutron sources that highlight the opportunities and the impact of laser-driven neutron sources. We hope that referee 2 is now convinced of the high quality of our R&D on laser-driven neutron sources.

“However, what is shown in this publication is likely as harmful to the translation of these techniques to application space as it is for pushing the field forward; as the results are not terribly convincing and discussion of the results lack rigorous error bars or consideration of how to improve the results and generalize the experimental setup such that others may use or build off of these techniques”

We acknowledge the comment of the referee that our work pushes the field forward. To mitigate the critique of referee 2, we have re-evaluated parts of the measurements, added a more rigorous evaluation of the uncertainties and added suggested improvements for a reduction in the error bars. The signal-to-noise ratio as well as the required number of laser pulses for a distinct application scenario is now thoroughly discussed. Improvements and perspectives are also given. We are confident that our results are not only useful for the field but also present a clear path towards applications of laser-driven neutron sources.

“The novelty and significance of this work is not well articulated. There have been other one-off demonstrations of short-pulse laser-driven NRS, NRI, and neutron radiography. This paper would be strengthened by either discussing how this latest work shows an improvement on other early work, or more clearly call attention to the fact of the versatility of the laser-driven neutron source that these can be done simultaneously or with just minor adjustments to the source or experimental geometry.”

In the revised manuscript, we explain the novelty and impact of our results in a clearer way. To our knowledge, epi-thermal or thermal laser-driven NRS, neutron radiography or neutron resonance imaging has not been published in peer reviewed journals. If reviewer 2 has additional information about publications that we are unaware of, we would be delighted if he/she could give us the corresponding references. According to our knowledge and our correspondence with other leading researchers in the field of laser-driven neutron sources, the only demonstrated applications are fast-neutron radiography [M.Roth et al, Phys. Rev. Lett. 2013] and an attempt on fast neutron spectroscopy by [Kishon et al., Nucl. Instruments and Methods 2019] which was limited by the small changes in the carbon cross section compared to the detector noise. We followed the advice of the referee and stressed in the revised manuscript that NRS or the NRI/neutron radiography with fast neutrons can be done in parallel.

Nonetheless, we know of two experimental groups that have tried to replicate our results with moderated neutrons since our experimental campaign in 2018 and 2020. But so far, none of these results have been published to our knowledge.

“In the introduction, it is claimed that the laser driven neutron sources could potentially compete with spallation or accelerator-based neutron sources such as LANSCE. However, few quantitative details are given on the requirements on the neutron flux and temporal pulse width and whether these new laser driven neutron sources can yet meet those requirements (and if not, how far are they?). For each of the NRS, NRI, radiography applications, how do the results obtained here compare to the state-of-the-art, what advantages or disadvantages does the laser-driven approach provide, and how much more development is needed to make them viable?”

We thank the reviewer for this advice and added a direct comparison to the LANSCE spallation source in the revised manuscript. In Figure 2, we have further added a direct comparison of the time-of-flight uncertainty between LANSCE and our LDNS design. It shows that a higher energy resolution can be obtained at LDNS, especially at energies in the 100 eV to 10's of keV range. This uncertainty is typically compensated by longer flight paths at LANSCE which reduces the usable neutron flux at the detector by a factor of 52 in comparison to the detector position at an LDNS. A direct comparison of the thermal flux was further added between LANSCE, the GELINA ToF facility and an LDNS. We discuss in the revised manuscript how an increase in repetition rate and laser energy will lead to powerful laser-driven neutron sources that can be compared to the thermal detector neutron flux of GELINA.

*“- This reviewer is relatively senior and familiar with the scientific literature in the high-intensity laser science field. However, I have never before come across the notation using the parenthesis (e.g., 103(9) J, or peak intensity of 2.0(3) * 10²⁰W/cm²). Is the number in parentheses meant to denote an alternate case or an error bar or an additional significant digit?”*

This notation refers to the last significant digits to their left outside the brackets, i.e. 1.15(5) means 1.15±0.05 while 1.15(10) means 1.15±0.10. We agree that this notation is not very common in our field but can be found in the particle acceleration community for instance. To avoid confusion, we use the ± notation in the revised manuscript.

“- The concept and characterization of the laser-driven epithermal neutron source is notable and is alone worthy of a publication, if further details could be given on how the experimental parameters scaled or influenced the resulting spectrum, and exactly which spectrum was best for which application. For example, in the Methods, it is mentioned the range of deuterated polystyrene foil targets used, and that three different converter designs were tested. However, there is no further description of which design ended up being used for each of the demonstrations shown in the paper.”

Based on the recommendation of the referee, we have added a detailed discussion of the converter characterization and modeling to the revised manuscript.

“Figure 1 (d) and (e) show the simulated moderator cross section in color bar units of $n/cm^2/proton$. But this is not explained at all in the paper; By the time the fast neutrons hit the moderator, the conversion from protons has already occurred in the converter foil. So what are (d) and (e) showing?? 1(b) appears to show an example of a fast neutron spectrum, but for what target and laser conditions, and/or if it is representative, it would be good to show error bars or a range of spectra achieved for this investigation. As this manuscript is primarily an experimental demonstration, specific geometries and setup are quite pertinent, so Fig. 1 of the supplementary information was very informative, and some combination of Fig. 1 in manuscript and Fig. 1 in supplementary would be helpful.”

We acknowledge the referee's hint to this ambiguity.

A PHITS simulation is performed by implementing a proton spectrum which was measured with Radiocromic Films (RCF) together with a Thomson parabola, and irradiating this proton distribution onto a converter material. The number of protons can vary from laser to laser and from shot to shot, therefore, to make an independent statement about the relative neutron flux, it is common in Monte Carlo simulations to normalize the results to numbers per incident proton. The total flux is calculated by multiplying these numbers with the integrated number of protons from a measurement.

We have replaced the single spectra with the average spectra for different data-sets and converters. It should be clear how the optimum target-converter combination was found. We think that the development of our optimized LDNS is now much clearer presented to the reader.

We followed the suggestion of the reviewer and added the figure of the experimental geometry to the manuscript.

„MCP = micro channel plate or multi channel plate?“

We thank the referee for the careful reading. We corrected the typo and wrote 'micro-channel plate' in the revised manuscript.

“What code was used for the NRS simulations? It is never explicitly said or details given about this simulation. Fig. 2(b) would be more convincing if the detector corrections and binning adjustments were applied to multiple resonances and could be shown to consistently match Breit-Wigner. This could also be extended to make a statement about the relative isotopic

concentrations, and/or if the measurement requires improvement to extract this, it should be stated.”

The simulations were performed with the Monte Carlo code PHITS (version 3.16) and the corresponding libraries. We now explicitly state this in the text.

Regarding Fig2(b), now Fig. 4b: The decreasing detector efficiency for higher energies and the smaller resonance sizes increase the statistical uncertainty for higher resonances. We tried to retrieve information on the concentration of the sample. Even though the Breit-Wigner fit seems to resemble the experimental data quite well, the statistics are the limiting factor for determining the concentration. This is owing to the fact that the ‘depth’ of the resonance must be measured for determining the concentration which requires small energy bins and neutron signal at the resonance position at the same time. It is explicitly stated in the discussion of the NRS measurement that our data cannot be used to identify concentrations but merely the presence of certain isotopes.

“-Is Fig.2(d) caption a typo? It appears that 1 eV neutrons should arrive later at the detector? Perhaps this needs more explanation.”

We have added a more detailed explanation of this phenomenon in the text. The fact that these neutrons arrive earlier than expected from the moderator-detector distance means that the moderated neutrons do not have their origin in the moderator. Instead, a fraction of fast neutrons are moderated in the borated polyethylene shielding and thus arrive earlier at the detector. These neutrons contribute to the background level (about 10% of the signal at the corresponding time).

“The paper could be improved wholly by citing error bars for all measurements. Discussion on how those error bars were derived can be placed in the Supplementary material.”

Error bars for all measurements have been added wherever applicable.

For the NRS measurement, single neutron events were detected. The uncertainty in this measurement comes from the statistical fluctuation in each energy bin and is evaluated in Fig. 4.c.

“An honest evaluation of the thermal neutron radiography technique would include a thickness and uniformity retrieval of the In sample as well.”

We added an evaluation of the inhomogeneous indium sample to the revised manuscript.

“Neutron Resonance Imaging is indeed a very powerful technique, however, it is not clear that the demonstration here is really a demonstration of NRI at all versus just neutron attenuation. The manuscript claims that the detected neutrons is influenced by the ^{113}Cd resonance, but there is no information given on what energy the resonance occurs at or how much attenuation is expected. A more convincing demonstration would have used ^{113}Cd covering a portion of the detector, and another material of a matching areal density (or material of very different resonance) to show differences in the attenuation.”

We agree with the referee and investigated a sample that contains indium and cadmium. We scanned through three different energy regions and compared the attenuation behind indium and cadmium. Two PHITS simulations have been added to make a direct comparison to the expected differences in attenuation.

Reviewer #3:

“Neutrons are a valuable tool for non-destructive material investigation. So far, most neutron applications have been limited to large-scale facilities such as nuclear research reactors, spallation sources, and accelerator-driven neutron sources. With the development of the laser system, laser-driven proton accelerators enable compact neutron source. Use for the laser-driven neutron source (LDNS) have been so far limited to neutron radiography. The authors report the attempt to use a LDNS to neutron resonance spectroscopy (NRS) and the neutron resonance imaging (NRI). The system is based on the laser which gives peak intensity of $2.0E20$ W/cm². Accelerated protons and deuterons are directed to a converter to generate neutrons via (p,n) and (d,n) reaction to yield neutrons. Neutrons from the converter enter the moderator made of polyethylene, which was optimized to give the maximum yield of 1~10 eV. The epithermal neutrons propagate through the sample, then detected afterward. The NRS measurement was successfully demonstrated using a tungsten sample. The transmitted spectrum clearly shows the sharp dips corresponding to the three tungsten isotopes, as well as impurity isotope Ta. Also, signal-to-background ratio 30:1 was obtained. Concerning the performance for thermal neutron radiography, neutron transmission measurement through the cadmium and indium sample demonstrates that the system can be used to measure distribution of the sample thickness, and the spatial position of the sample was determined with an accuracy of 2 mm.”

We thank the referee for the careful reading of our manuscript.

Points of concern:

“Then the authors attempted the NRI measurement. However, I think the measurement did not demonstrate the scope of NRI. As I understand, it is the approach to identify the material from the resonance position and to find their distribution (location) over the sample as well as its amount.”

We agree with the referee that identifying the material and finding their location across the sample is the ultimate scope of NRI measurements. A determination of the amount can be an additional benefit of NRI but is not intrinsically necessary.

However, we disagree with the referee in the point that we have not done a variant of NRI. We have identified the position of cadmium across the sample by exploiting the large resonance of cadmium in the thermal region. To make clear that we measured different energy ranges, we have included a more detailed discussion of NRI in the revised manuscript by scanning from 1 meV to 1 eV in three steps. This a variant of the NRI technique revealed gradual change in attenuation for resonant regions are investigated in comparison to non-resonant regions. Our results are in good agreement with the expectations.

“I admit that the strong absorption of Cd113 is associated with the resonance at the thermal energy region. But this is a very special case found in Cd113. In general, NRI should be adopted to more general material who has resonances at higher energies. Actually, the present NRS experiment shows the resonances in the neutron energy range of 1 -10 eV, which is larger than the analysis given in FIG.4, by which the authors claim the demonstration of NRI. Thus, it is difficult to agree that the authors proved the applicability of the NRI using LDNS. I believe analysis of different material needs to be done to claim the success of NRI. “

We agree that different materials and higher energies have to be investigated. For this reason, we compare the attenuation in cadmium with the attenuation in indium for different energy ranges. From the changes in the attenuation, cadmium can be clearly identified. Cadmium is used for this proof-of-principle experiment because of its resonance at low energies. However, it is less than a special case. ²³⁹Pu has a similar low-energy resonance at 300 meV and another resonance at 1.05 eV. This makes it directly comparable and we added a short discussion in the revised manuscript.

We understand the referee’s wish to see an NRI measurement at higher energies as more resonances are in this range. Based on the findings with our optimized LDNS, we extrapolate and provide estimates how many how many shots are needed to investigate resonances at higher energies (Table 1 and Fig.7). Realistic application scenarios with high-repetition rate lasers are provided showing the application potential of LDNS.

REVIEWERS' COMMENTS

Reviewer #3 (Remarks to the Author):

In the revised manuscript, the experimental methods and results are explained in more detail. The new figures help the readers to reach clear understanding. The style of the article is significantly improved. Also, the number of laser-shot given in the measurements is addressed, which will significantly impress the readers.

To answer my question concerning the achievement of the NRI measurement, the authors modified the manuscript largely. I made the comment that Cd113 is a special nuclide having large cross section in thermal energy region, and the measurement cannot be a direct proof of the availability of NRI. The authors answered my question by showing the neutron-energy dependence of the imaging in Fig.6. The figure becomes more convincing than Fig.4 of the first version. Also, the authors showed the necessary number of laser-shot and a required measurement time is examined in each nuclide. Considering all these efforts and new presentation of the data, now I can admit that the reported result is the first demonstration of the NRI using LDNS.

With the revised version of the manuscript, I recommend this article to be published in Nature Communications.

Response to the reviewers:

Reviewer 3:

In the revised manuscript, the experimental methods and results are explained in more detail. The new figures help the readers to reach clear understanding. The style of the article is significantly improved. Also, the number of laser-shot given in the measurements is addressed, which will significantly impress the readers.

To answer my question concerning the achievement of the NRI measurement, the authors modified the manuscript largely. I made the comment that Cd113 is a special nuclide having large cross section in thermal energy region, and the measurement cannot be a direct proof of the availability of NRI. The authors answered my question by showing the neutron-energy dependence of the imaging in Fig.6. The figure becomes more convincing than Fig.4 of the first version. Also, the authors showed the necessary number of laser-shot and a required measurement time is examined in each nuclide. Considering all these efforts and new presentation of the data, now I can admit that the reported result is the first demonstration of the NRI using LDNS.

With the revised version of the manuscript, I recommend this article to be published in Nature Communications.

We thank the reviewer for his/her revised view of our manuscript and as no comments for improvement were made we did not have to change the manuscript.